# Diabetic Dermopathies as Predictive Markers of Cardiovascular and Renal Complications: A Narrative Review

**DOI:** 10.3390/jcm14217719

**Published:** 2025-10-30

**Authors:** Madalina Marinescu, Gina Eosefina Botnariu, Dan Vâță, Adriana-Ionela Patrascu, Doinița Temelie-Olinici, Mădălina Mocanu, Ioana Halip, Ioana Adriana Popescu, Dragoș Florin Gheuca-Solovastru, Laura Gheuca-Solovastru

**Affiliations:** 1Faculty of Medicine “Grigore T. Popa”, University of Medicine and Pharmacy, 700111 Iași, Romania; dr.madalinamarinescu@gmail.com (M.M.); patrascuai@yahoo.com (A.-I.P.); doinitzaganceanu@yahoo.com (D.T.-O.); drmadalinamocanu@yahoo.com (M.M.); ioana-alina.grajdeanu@umfiasi.ro (I.H.); oana.manolache@yahoo.com (I.A.P.); dragosflorin13@yahoo.com (D.F.G.-S.); lsolovastru13@yahoo.com (L.G.-S.); 2Clinic of Dermatology-Venereology, “Sf. Spiridon” Emergency County Clinical Hospital, 700111 Iași, Romania

**Keywords:** diabetic dermopathy, cutaneous manifestations of diabetes, skin lesions AND diabetes, necrobiosis lipoidica, scleredema diabeticorum, bullosis diabeticorum, eruptive xanthomas, cutaneous biomarkers AND systemic complications

## Abstract

**Background**: Cutaneous manifestations are frequent in diabetes mellitus and may reflect systemic vascular injury. Among them, diabetic dermopathy, necrobiosis lipoidica, scleredema diabeticorum, bullosis diabeticorum, and eruptive xanthomas are clinically significant. Aim: To synthesize current evidence on the associative and potential predictive role of diabetic dermopathies as non-invasive indicators of cardiovascular and renal complications. **Methods**: A narrative review of studies published between 2010 and 2023 in PubMed, Scopus, and Web of Science was conducted, focusing on links between specific dermopathies and systemic outcomes. **Results**: Diabetic dermopathy shows consistent associations with microvascular complications such as retinopathy and nephropathy. Necrobiosis lipoidica and scleredema correlate with macrovascular disease and metabolic syndrome, whereas eruptive xanthomas indicate severe dyslipidemia and heightened cardiovascular risk. Evidence is predominantly cross-sectional, with limited sample sizes and heterogeneous diagnostic criteria. **Conclusions**: Diabetic dermopathies represent emerging clinical indicators of systemic vascular and metabolic burden. Their potential prognostic value supports the integration of dermatological assessment into comprehensive diabetes care. However, due to methodological limitations, these findings should be interpreted as associative rather than causal, and prospective studies are warranted to confirm their predictive significance.

## 1. Introduction

Diabetes mellitus (DM) is one of the most significant global health challenges of the 21st century. According to the International Diabetes Federation (IDF), more than 537 million adults were living with diabetes in 2021, and this number is projected to rise to 643 million by 2030 and 783 million by 2045 [1]. The World Health Organization (WHO) has recognized diabetes as a major contributor to premature mortality and disability worldwide, with an estimated 6.7 million deaths attributed to the disease in 2021 alone [2]. Beyond mortality, diabetes imposes an enormous economic burden on healthcare systems, accounting for approximately 966 billion USD in global health expenditures in 2021, a figure expected to continue rising steeply [3].

The morbidity associated with diabetes is primarily linked to its complications, which are traditionally divided into microvascular and macrovascular categories. Microvascular complications include diabetic retinopathy, nephropathy, and neuropathy, while macrovascular complications encompass coronary artery disease, cerebrovascular disease, and peripheral arterial disease [4,5]. These complications are the leading causes of disability and reduced quality of life in diabetic patients [6]. While laboratory markers and imaging modalities remain the cornerstone of diagnosing and monitoring such complications, the cutaneous manifestations of diabetes provide a unique, easily accessible clinical window into the systemic burden of the disease.

Cutaneous manifestations occur in more than 30% of diabetic patients and can often precede or accompany systemic complications [7]. The skin, as the largest and most visible organ, reflects systemic metabolic and vascular changes. Historically, these manifestations were considered of secondary importance or cosmetic. However, increasing evidence demonstrates that dermatological changes not only mirror underlying systemic dysfunction but may also serve as early predictors of microvascular and macrovascular disease [8,9].

A variety of cutaneous conditions have been described in diabetes, ranging from common lesions such as diabetic dermopathy to rarer but highly specific disorders such as necrobiosis lipoidica, scleredema diabeticorum, bullosis diabeticorum, and eruptive xanthomas [10,11,12]. Diabetic dermopathy, often referred to as “shin spots,” is characterized by atrophic, hyperpigmented macules on the pretibial area and occurs in up to 50% of patients with long-standing diabetes [13]. Necrobiosis lipoidica, affecting 0.3–1.2% of diabetic patients, manifests as yellow-brown plaques with telangiectasias and central atrophy, frequently complicated by ulceration [14]. Scleredema diabeticorum presents with thickening and induration of the posterior neck and upper back, predominantly in obese patients with poor glycemic control [15]. Bullosis diabeticorum, although rare, is notable for its sudden appearance of large, non-inflammatory bullae on acral sites [16]. Eruptive xanthomas occur in the context of severe hypertriglyceridemia, presenting as yellow-red papules on extensor surfaces and buttocks, and indicate significant dyslipidemia [17].

The pathophysiological mechanisms linking diabetes to its cutaneous manifestations are complex and multifactorial. Chronic hyperglycemia induces non-enzymatic glycation of proteins, resulting in the accumulation of advanced glycation end-products (AGEs), which contribute to endothelial dysfunction, oxidative stress, and impaired microcirculation [18]. Microangiopathy, a hallmark of diabetic complications, is characterized by thickening of capillary basement membranes and reduced tissue perfusion, processes that also affect the skin [8]. Inflammatory pathways play a critical role, with elevated cytokine levels (e.g., TNF-α, IL-6) and chronic low-grade systemic inflammation promoting both cutaneous and systemic vascular damage [19]. Insulin resistance and dyslipidemia further exacerbate these processes, especially in conditions like scleredema diabeticorum and eruptive xanthomas [20].

Importantly, multiple studies have demonstrated correlations between cutaneous lesions and systemic complications. Diabetic dermopathy has been associated with higher prevalence of retinopathy, nephropathy, and neuropathy [21]. Necrobiosis lipoidica has been linked with peripheral arterial disease and coronary artery disease [21]. Scleredema diabeticorum correlates with metabolic syndrome, insulin resistance, and left ventricular dysfunction [22]. Eruptive xanthomas are widely recognized as external markers of severe hypertriglyceridemia and atherosclerotic cardiovascular risk [7]. These associations underscore the potential role of skin manifestations as non-invasive, cost-effective biomarkers for systemic complications in diabetes.

The recognition of cutaneous manifestations has important clinical implications. Dermatologists may be the first specialists to detect these lesions, thereby identifying patients at higher risk for systemic complications. For diabetologists and primary care physicians, awareness of these signs may prompt timely cardiovascular and renal evaluation, leading to earlier interventions. This highlights the importance of interdisciplinary collaboration in diabetes management [23].

Despite growing interest in the subject, current evidence is limited by small sample sizes, heterogeneity in diagnostic criteria, and lack of large-scale prospective studies. Furthermore, dermatological findings are often underreported in clinical practice and rarely included in risk stratification models [24,25]. Emerging diagnostic tools, such as dermoscopy, skin autofluorescence, and high-frequency ultrasound, may enhance the diagnostic accuracy and prognostic value of these lesions [26]. In addition, novel therapeutic agents for diabetes, such as GLP-1 receptor agonists and SGLT2 inhibitors, may have unrecognized effects on cutaneous manifestations, an area requiring further research [27].

In this context, the present review aims to provide a comprehensive synthesis of the literature on diabetic dermopathies, focusing on their prevalence, pathophysiology, clinical spectrum, and prognostic implications. Special emphasis is placed on their role as predictive markers of cardiovascular and renal complications, with the goal of highlighting their potential utility in risk stratification and integrated diabetes care.

Despite growing evidence linking diabetic dermopathies to systemic vascular injury, no previous review has specifically synthesized their potential predictive value for cardiovascular and renal complications. Existing studies are often limited by small sample sizes, cross-sectional designs, and heterogeneous diagnostic criteria, leading to fragmented knowledge and inconsistent conclusions. Thus, a comprehensive narrative synthesis focusing on the associative and emerging prognostic role of diabetic dermopathies is still lacking. The present review aims to fill this gap by critically summarizing current evidence on the prevalence, pathophysiological mechanisms, and potential value of cutaneous manifestations as non-invasive indicators of systemic vascular complications in diabetes.

## 2. Material and Methods

This article is designed as a narrative review, aiming to provide an integrative synthesis of the available evidence on diabetic dermopathies and their associations with cardiovascular and renal complications. Unlike systematic reviews, a narrative approach focuses on summarizing and interpreting the literature in a descriptive and clinically oriented manner, without following PRISMA guidelines or performing quantitative meta-analysis.”

### 2.1. Literature Search Strategy

A comprehensive literature search was carried out in three major electronic databases: PubMed/MEDLINE, Scopus, and Web of Science. The search covered the period between January 2010 and December 2023. Key search terms included: “*diabetic dermopathy*,” “*cutaneous manifestations of diabetes*,” “*skin lesions AND diabetes*,” “*necrobiosis lipoidica*,” “*scleredema diabeticorum*,” “*bullosis diabeticorum*,” “*eruptive xanthomas*,” and “*cutaneous biomarkers AND systemic complications*.” Boolean operators (AND, OR) were applied to combine terms and maximize retrieval.

To ensure sensitivity, both MeSH terms and free-text keywords were included. For example, in PubMed, the following strategy was used:(“diabetic dermopathy”[MeSH] OR “diabetic skin lesions”) AND (“microangiopathy” OR “retinopathy” OR “nephropathy” OR “neuropathy”)(“necrobiosis lipoidica” OR “scleredema diabeticorum” OR “bullosis diabeticorum” OR “eruptive xanthomas”) AND (“diabetes mellitus”[MeSH])

Additionally, reference lists of selected papers were manually screened to identify further relevant publications not captured in the electronic search [27,28].

### 2.2. Inclusion and Exclusion Criteria

Articles were selected based on predefined eligibility criteria.

Inclusion criteria:

Studies involving patients with type 1 or type 2 diabetes mellitus.Observational designs (cross-sectional, case–control, cohort), clinical trials, systematic reviews, and meta-analyses.Explicit evaluation of cutaneous manifestations and their correlation with microvascular or macrovascular complications.Publications in English language.Full-text availability.

Exclusion criteria:

Conference abstracts, letters to the editor, and narrative reports without sufficient methodological detail.Experimental animal studies or in vitro research without clinical correlation.Non-English articles.Papers focusing exclusively on non-diabetic dermatological disorders.

### 2.3. Data Extraction and Synthesis

Two independent reviewers (conceptually represented in this review) screened titles and abstracts to assess eligibility. Full texts were then evaluated in detail. Extracted data included:First author and year of publicationCountry and study settingSample size and patient demographicsType of dermopathy investigatedMethod of dermatological diagnosis (clinical vs. histological)Systemic outcomes measured (e.g., presence of retinopathy, nephropathy, cardiovascular disease)Main results and conclusions

Due to significant heterogeneity across studies in terms of design, diagnostic criteria, and outcomes, meta-analysis was not feasible. Instead, results were synthesized narratively and grouped according to the specific dermopathy studied (diabetic dermopathy, necrobiosis lipoidica, scleredema, bullosis diabeticorum, eruptive xanthomas). Where possible, prevalence rates and correlation coefficients were highlighted [28,29,30].

To ensure methodological rigor, the reference list was carefully reviewed to avoid duplication and to maintain a balanced representation between primary research (observational and cohort studies) and secondary narrative reviews. Priority was given to original studies providing primary data on associations between dermopathies and systemic complications.

### 2.4. Study Selection and Data Summary

Given the narrative nature of this review, studies were selected based on relevance, clinical content, and contribution to understanding the relationship between diabetic dermopathies and systemic complications. No formal risk-of-bias tool (e.g., Newcastle–Ottawa Scale) was applied, as the purpose was descriptive synthesis rather than quantitative appraisal. To enhance transparency, Table 1 provides an overview of representative studies, including study design, dermopathy investigated, sample size, and main associations.

### 2.5. Quality Assessment

Although formal risk-of-bias assessment tools (such as Newcastle–Ottawa Scale or Cochrane risk-of-bias tool) were not systematically applied, the methodological rigor of each study was considered in interpretation. Studies with larger sample sizes, objective diagnostic criteria, and appropriate statistical adjustments were given greater weight. Small case series and anecdotal reports were included for completeness but interpreted cautiously [28,30].

### 2.6. Comparison with Existing Reviews

A number of systematic reviews on diabetic skin complications exist [31,32]. However, most focus broadly on cutaneous disorders without emphasis on their predictive value for cardiovascular and renal outcomes. Our approach specifically targeted the intersection between dermatology and internal medicine, providing a multidisciplinary synthesis that complements previous reviews.

### 2.7. Rationale for Narrative Approach

The narrative method was considered most appropriate because of the following:The relative scarcity of large prospective cohort studies.The diversity of dermopathies (ranging from benign to rare, severe lesions).Heterogeneity in reported outcomes (e.g., microvascular vs. macrovascular vs. metabolic syndrome endpoints).

Narrative review methodology allowed the integration of heterogeneous evidence, contextual interpretation, and critical appraisal of clinical relevance.

## 3. Results

The analysis of the reviewed literature reveals a wide clinical spectrum of cutaneous manifestations in diabetes mellitus, with significant variability in prevalence, severity, and correlation with systemic complications.

### 3.1. Diabetic Dermopathy

Diabetic dermopathy is the most frequent cutaneous manifestation of diabetes, described in up to 50% of patients with long-standing type 2 diabetes [8]. These lesions typically present as small, round, atrophic, brownish macules located on the anterior shins. Histopathologically, they show basal cell hyperpigmentation, hemosiderin deposition, and dermal fibrosis [8].

Multiple studies confirm a strong correlation between dermopathy and microvascular complications. Mirhoseini et al. reported that patients with dermopathy were significantly more likely to have diabetic retinopathy and nephropathy compared to those without lesions [21]. Similarly, Mirhoseini et al. found that the number and severity of dermopathy lesions correlated with the degree of retinopathy and albuminuria [21]. This suggests that dermopathy is not merely a cosmetic marker but a visible sign of systemic microangiopathy.

### 3.2. Necrobiosis Lipoidica

Necrobiosis lipoidica diabeticorum (NLD) is relatively rare, with a prevalence between 0.3 and 1.2% [11]. Clinically, it presents as yellow-brown plaques with central atrophy and telangiectasias, predominantly located on the shins. Lesions may ulcerate in up to 35% of cases, leading to chronic wounds [8].

NLD has been linked to both microvascular and macrovascular complications. A study by Boulton et al. demonstrated that patients with NLD exhibited higher prevalence of peripheral arterial disease [11]. Other investigations have reported associations with coronary artery disease, suggesting that NLD reflects systemic atherosclerosis beyond cutaneous involvement [8]. Histopathology reveals granulomatous inflammation, collagen degeneration, and thickened blood vessels, supporting its role as a marker of vascular injury.

### 3.3. Scleredema Diabeticorum

Scleredema diabeticorum is a rare but clinically significant disorder, characterized by non-pitting induration of the posterior neck and upper back [12]. Prevalence is difficult to establish, but studies estimate rates up to 2.5% in obese diabetic populations. Rho et al. reported a strong correlation between scleredema and metabolic syndrome, including hypertension, dyslipidemia, and insulin resistance [19].

Cardiovascular associations are particularly relevant. A cross-sectional study by Rho et al. revealed that scleredema was associated with left ventricular hypertrophy and diastolic dysfunction [19]. This suggests that scleredema may be an external marker of subclinical cardiac disease in diabetes.

### 3.4. Bullosis Diabeticorum

Bullosis diabeticorum is an uncommon blistering condition characterized by spontaneous, painless bullae on acral sites such as feet and hands. Though rare (<0.5% of diabetic patients), its recognition is important [13]. El Fekih et al. described 10 patients with bullosis diabeticorum, all of whom had advanced microangiopathy [13]. The condition is thought to result from microvascular ischemia leading to dermo-epidermal separation.

While bullosis diabeticorum itself is self-limiting, its presence indicates severe systemic vascular damage and long-standing diabetes with poor glycemic control. It has been proposed as a cutaneous red flag for advanced microvascular complications [7].

### 3.5. Eruptive Xanthomas

Eruptive xanthomas occur in the context of severe hypertriglyceridemia, often exceeding 2000 mg/dL. Clinically, they present as crops of yellow-red papules on extensor surfaces and buttocks [14]. In diabetic patients, eruptive xanthomas are frequently associated with poorly controlled disease and metabolic syndrome [25].

The systemic significance lies in their correlation with acute pancreatitis and high cardiovascular risk. Kanitakis highlighted that eruptive xanthomas represent a visible sign of severe atherogenic dyslipidemia, necessitating urgent metabolic control [14]. Armstrong et al. linked their occurrence to premature coronary artery disease in young diabetic adults [20].

### 3.6. Integrated Correlations with Systemic Complications

Across dermopathies, several patterns emerge:Microangiopathy: Dermopathy and bullosis diabeticorum correlate strongly with retinopathy, nephropathy, and neuropathy [21].Macroangiopathy: Necrobiosis lipoidica and scleredema show stronger associations with peripheral arterial disease, coronary artery disease, and metabolic syndrome [8,11].Dyslipidemia: Eruptive xanthomas are external markers of profound lipid abnormalities that contribute to accelerated atherosclerosis [14,20].

Overall, cutaneous signs reflect a shared pathophysiology of vascular injury, involving endothelial dysfunction, chronic inflammation, and glycation end-products. Recognition of these patterns provides clinicians with non-invasive diagnostic cues.

To facilitate clinical interpretation and practical application, Table 2 provides a concise summary of the major diabetic dermopathies, including their predominant association with diabetes type, key clinical features, differential diagnoses, and recommended management strategies.

### 3.7. Pathophysiological Considerations

The pathophysiological mechanisms linking diabetic dermopathies to systemic complications are complex and multifactorial. Chronic hyperglycemia induces endothelial dysfunction, thickening of capillary basement membranes, and accumulation of advanced glycation end-products (AGEs) in both skin and internal organs. Inflammatory mediators such as TNF-α and IL-6 amplify oxidative stress and vascular injury, while dyslipidemia accelerates atherogenesis. The dermal microangiopathy observed in diabetic dermopathy parallels retinal microangiopathy in retinopathy and glomerular changes in nephropathy. Necrobiosis lipoidica demonstrates granulomatous inflammation and collagen degeneration similar to systemic atherosclerosis. Scleredema reflects excessive glycosylation of collagen and extracellular matrix proteins, leading to thickened connective tissue analogous to myocardial fibrosis. These pathophysiological overlaps suggest that dermopathies reflect systemic vascular alterations and may act as cutaneous indicators of underlying vascular disease, although current evidence remains primarily associative.

### 3.8. Other Cutaneous Conditions Associated with Diabetes Mellitus

Beyond the classical dermopathies with potential predictive value, numerous other dermatologic manifestations are frequently encountered in diabetic patients and carry significant clinical implications.

Pruritus: Generalized or localized pruritus, particularly in the genital or lower limb regions, is common and may result from xerosis, neuropathy, or secondary infections. Optimal glycemic control, emollients, and treatment of underlying causes are essential.Acanthosis nigricans: Characterized by hyperpigmented, velvety plaques in intertriginous areas (neck, axillae), acanthosis nigricans is a cutaneous marker of insulin resistance and metabolic syndrome, often preceding diabetes onset.Granuloma annulare: The localized form, presenting as annular papules on hands and feet, has been reported more frequently in diabetic patients, though the association is not fully understood.Skin reactions related to medical device use: Repeated insulin injections, continuous glucose monitoring sensors, and insulin pumps can cause lipohypertrophy, lipoatrophy, or allergic contact dermatitis due to adhesives or preservatives. Proper site rotation and hypoallergenic materials can minimize these reactions.Diabetic foot ulcers: Resulting from neuropathy, ischemia, and infection, these lesions are a major cause of morbidity and amputation. Regular foot examination and multidisciplinary management are critical for prevention and treatment.Recurrent cutaneous infections: Diabetes predisposes to bacterial (furuncles, carbuncles), fungal (candidiasis, dermatophytosis), and viral infections due to impaired immune response and hyperglycemia.Other dermatoses: Conditions such as eruptive xanthomas, skin tags, and xerosis are associated not only with diabetes but also with obesity, dyslipidemia, chronic kidney disease, and hypothyroidism, reflecting the multisystem metabolic burden.

These conditions (Figure 1), although not primarily predictive markers, contribute to disease burden, affect quality of life, and may serve as clinical indicators of poor metabolic control or comorbidities.

### 3.9. Clinical Implications

The practical implication is that dermatological examination should be integrated into routine diabetic care. Patients presenting with dermopathy or scleredema should undergo targeted screening for retinopathy, nephropathy, and cardiovascular risk factors [7]. Similarly, eruptive xanthomas should trigger immediate lipid evaluation and management to prevent life-threatening pancreatitis and cardiovascular events [14].

To minimize redundancy and facilitate critical comparison across studies, we summarize key findings in Table 3, including approximate prevalence, predominant systemic associations, a qualitative level-of-evidence rating, and documented inconsistencies or negative findings.

## 4. Discussion

Unlike previous reviews that primarily cataloged cutaneous manifestations, this work focuses on the emerging predictive value of diabetic dermopathies and provides a comparative synthesis of evidence strength and contradictions, highlighting knowledge gaps relevant for future prognostic research. The findings of this review highlight the clinical and prognostic relevance of diabetic dermopathies as potential non-invasive indicators of systemic complications, particularly those affecting the cardiovascular and renal systems. Historically underappreciated and often regarded as cosmetic concerns, these cutaneous manifestations are increasingly recognized as cutaneous correlates of microvascular and macrovascular injury. In the following discussion, we integrate evidence from observational studies, pathophysiological insights, and clinical practice to contextualize the emerging predictive role of dermopathies in diabetes management.

### 4.1. Correlation with Microvascular Complications

Several studies have reported associations between diabetic dermopathy and microvascular complications such as retinopathy and nephropathy [21]. Although these findings are consistent across multiple cohorts, the predominance of cross-sectional designs limits causal inference. Dermopathy should therefore be considered an emerging clinical indicator rather than a validated biomarker. Bullosis diabeticorum, though rare, is frequently seen in patients with advanced microangiopathy and long-standing diabetes, suggesting it may act as a clinical red flag for severe microvascular compromise [13]. Recognition of these lesions should prompt clinicians to investigate other microvascular complications, including retinopathy, nephropathy, and neuropathy. However, most of these studies are cross-sectional and limited by small sample sizes, which restricts the ability to infer causality or establish predictive value.

### 4.2. Correlation with Macrovascular Complications

Necrobiosis lipoidica and scleredema diabeticorum appear more closely related to macrovascular disease and metabolic syndrome. NLD has been associated with peripheral arterial disease and coronary artery disease [8,11], while scleredema correlates with obesity, insulin resistance, and left ventricular dysfunction [12,19]. These associations are biologically plausible but remain based on small-scale, cross-sectional studies. Further longitudinal research is required to determine whether these dermopathies are independent predictors or simply reflect the broader metabolic burden. Reported associations are biologically plausible but should be interpreted cautiously due to heterogeneity in diagnostic criteria and limited longitudinal data.

### 4.3. Clinical Relevance of Eruptive Xanthomas

Eruptive xanthomas illustrate a direct link between cutaneous signs and metabolic derangements. These papular lesions indicate profound hypertriglyceridemia, often exceeding 2000 mg/dL, which carries immediate risk for pancreatitis and long-term cardiovascular morbidity [14]. In diabetic patients, their presence should trigger urgent lipid evaluation and therapeutic intervention. While the pathophysiological connection is well established, their role as predictive markers of cardiovascular outcomes remains to be confirmed.

### 4.4. Integration into Multidisciplinary Care

Dermatological findings can serve as adjunctive clinical information in diabetes management. Dermatologists may be the first to identify lesions that suggest increased systemic risk, while diabetologists, nephrologists, and cardiologists can interpret these signs within broader risk stratification models. However, current evidence supports dermopathies as associative markers rather than established biomarkers. Incorporating skin examination into clinical assessment may facilitate earlier detection of high-risk patients but should be interpreted alongside traditional risk factors and laboratory data. Given these methodological constraints, dermopathies should be regarded as adjunctive clinical indicators rather than established biomarkers until stronger evidence emerges.

### 4.5. Implications for Research

Despite the growing body of literature, the current evidence base is limited by small sample sizes, heterogeneity in diagnostic definitions, and reliance on observational designs [22]. Large-scale prospective cohort studies are needed to determine whether dermopathies independently predict cardiovascular and renal outcomes after adjusting for confounders. Emerging diagnostic tools such as skin autofluorescence, high-frequency ultrasound, and optical coherence tomography may improve the evaluation of cutaneous microangiopathy and its systemic implications [9,23]. Additionally, the effects of modern antidiabetic agents (e.g., GLP-1 receptor agonists, SGLT2 inhibitors) on cutaneous manifestations remain largely unexplored [32]. Investigating whether systemic improvements translate into changes in dermopathies could further clarify their prognostic value.

Shortcomings of the Existing Evidence

Despite increasing recognition of the association between diabetic dermopathies and systemic complications, the current body of evidence remains limited in several important aspects:Small sample sizes and single-center designs: Most studies are observational and include fewer than 150 participants, limiting statistical power and generalizability.Cross-sectional methodology: The absence of longitudinal follow-up precludes establishing temporal or causal relationships between cutaneous signs and systemic outcomes.Diagnostic variability: Definitions of dermopathies differ between studies (clinical vs. histopathological), creating heterogeneity and complicating direct comparison.Lack of standardized assessment tools: No validated scoring systems or unified diagnostic criteria exist, leading to potential misclassification.Publication bias: Positive associations are more likely to be published, while negative or null results are underrepresented.Confounding factors: Many studies fail to adjust for disease duration, glycemic control, or comorbidities, which may partly explain observed associations.Therapeutic data scarcity: Although recent agents such as GLP-1 receptor agonists and SGLT2 inhibitors show systemic benefits, no robust clinical trials have specifically examined their effect on dermopathies; available evidence is anecdotal or observational.

Future research should focus on prospective, multicenter studies using standardized diagnostic criteria, controlling for confounders, and integrating objective skin biomarkers to validate the prognostic role of these cutaneous manifestations.

### 4.6. Limitations of Current Evidence

While numerous studies report associations between diabetic dermopathies and systemic complications, the overall quality of evidence remains limited. Most available data derive from small, single-center cohorts and cross-sectional designs, which preclude causal inference and temporal validation of findings [22].

There is also considerable heterogeneity in diagnostic criteria across studies. For example, definitions of necrobiosis lipoidica and scleredema diabeticorum vary between clinical and histopathological descriptions, leading to inconsistent classification and variable prevalence estimates. Similarly, the diagnosis of diabetic dermopathy is often made clinically, with limited histological confirmation, which may result in misclassification [6,10].

Another major limitation is publication bias. Studies demonstrating positive or significant associations are more likely to be published, while negative or null results are underrepresented in the literature. This bias may overestimate the strength of associations reported in reviews such as the present one.

Furthermore, few studies adjust for confounding factors such as age, diabetes duration, glycemic control, or coexisting comorbidities, which may account for part of the observed correlations. The lack of standardized outcome measures and variability in study design complicate direct comparison and meta-analytic synthesis.

Therefore, while dermopathies appear to be promising associative markers, current evidence should be interpreted with caution. Prospective, multicenter studies with standardized diagnostic criteria and adjustment for confounders are required to confirm their independent predictive value and to establish their potential role in clinical risk stratification.

Although we systematically removed duplicate and overlapping references and prioritized primary data sources, the available literature remains limited, with many small observational studies and few large-scale prospective analyses. As such, some reliance on secondary narrative reviews was unavoidable due to the scarcity of high-quality primary data.

### 4.7. Limitations of Methodology

This review has several important limitations. First, the narrative design, while appropriate for integrating heterogeneous evidence, lacks the methodological rigor of systematic reviews or meta-analyses. The inclusion of observational studies with variable quality, small sample sizes, and inconsistent diagnostic definitions limits the strength of conclusions. Second, most data derive from cross-sectional designs, precluding causal inference. Third, publication and selection biases cannot be excluded, as positive associations are more likely to be reported. Additionally, the restriction to English-language publications may have excluded relevant data. Finally, the absence of standardized diagnostic criteria for several dermopathies complicates direct comparisons and hinders quantitative synthesis.

Despite these limitations, the review provides a comprehensive and integrative overview of diabetic dermopathies and their associations with systemic complications, highlighting key gaps and directions for future prospective research.

Nevertheless, despite these limitations, the methodology ensured a comprehensive and clinically oriented overview of the available literature on diabetic dermopathies as markers of systemic complications [33,34].

### 4.8. Toward Clinical Integration

For dermopathies to be recognized as formal clinical tools, standardized diagnostic criteria and validation in predictive models are essential. Until stronger evidence emerges, these conditions should be regarded as potential cutaneous indicators that complement existing risk assessments. Cutaneous examination remains an accessible, cost-effective adjunct that may enhance early detection of high-risk individuals, particularly in resource-limited settings.

### 4.9. Summary of Discussion

In summary, diabetic dermopathies are promising yet unvalidated indicators of systemic vascular injury. Dermopathy and bullosis diabeticorum show associations with microvascular complications, necrobiosis lipoidica and scleredema with macrovascular disease, and eruptive xanthomas with severe dyslipidemia and cardiovascular risk. Recognizing these patterns may facilitate early risk stratification and foster interdisciplinary collaboration in diabetes care.

## 5. Conclusions

Cutaneous manifestations remain a clinically relevant but often underrecognized component of diabetes mellitus. Among them, diabetic dermopathy, necrobiosis lipoidica, scleredema diabeticorum, bullosis diabeticorum, and eruptive xanthomas appear to be associated with systemic vascular and metabolic dysfunction, including microvascular complications (retinopathy, nephropathy), macrovascular disease, and severe dyslipidemia.

While these associations are biologically plausible and consistently reported across multiple observational studies, the evidence base is predominantly cross-sectional, with small, heterogeneous cohorts and variable diagnostic definitions. As a result, current data support an associative rather than causal or predictive interpretation.

Dermopathies may serve as potential non-invasive clinical indicators to raise suspicion of systemic vascular injury and to prompt targeted screening. Their recognition could add value to integrated diabetic care, particularly in low-resource settings where access to advanced imaging is limited.

However, before these manifestations can be considered validated prognostic biomarkers, further research is needed. Future prospective, multicenter studies employing standardized diagnostic frameworks, objective assessment tools, and adjustment for confounding factors are essential to confirm whether dermopathies have independent predictive utility for cardiovascular and renal outcomes.

Until such data become available, diabetic dermopathies should be viewed as emerging associative markers and visual clinical clues, complementing—but not replacing—established risk factors in comprehensive patient evaluation.

## Figures and Tables

**Figure 1 jcm-14-07719-f001:**
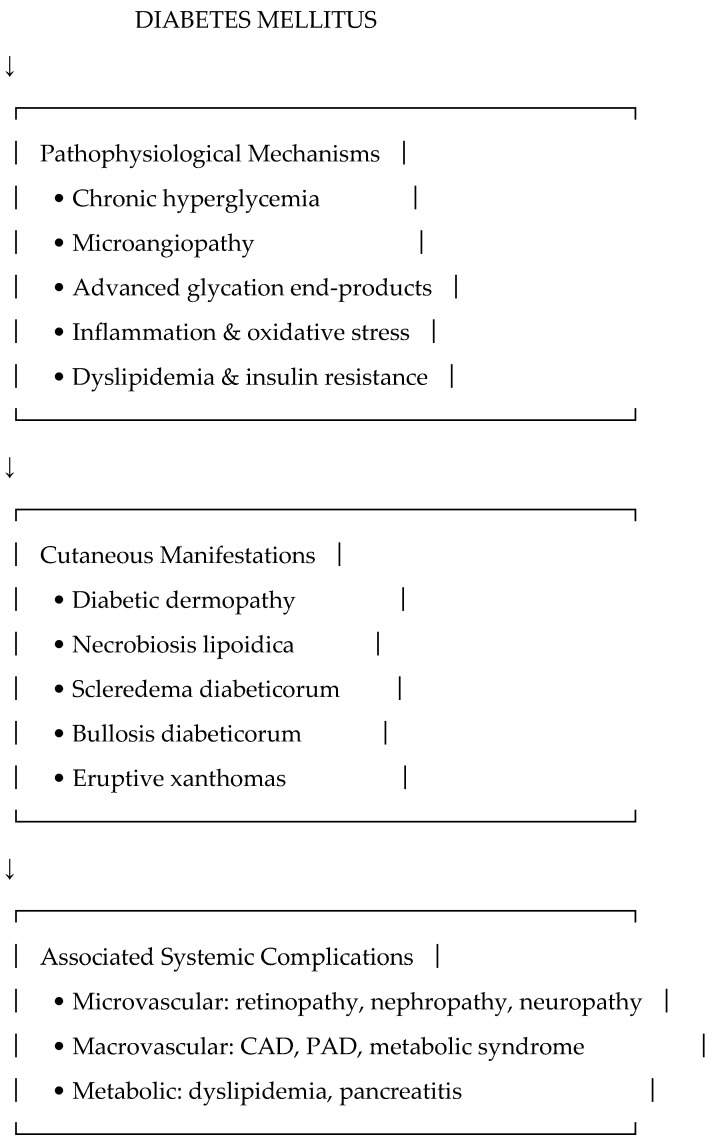
Conceptual framework linking diabetic dermopathies with pathophysiological mechanisms and systemic complications.

**Table 1 jcm-14-07719-t001:** Summary of representative studies exploring associations between diabetic dermopathies and systemic complications.

Author (Year)	Study Design	Cutaneous Manifestation	Sample Size	Main Systemic Association
Mirhoseini et al. (2016) [21]	Cross-sectional	Diabetic dermopathy	120	Strong association with diabetic retinopathy and nephropathy in type 2 DM
Demirseren et al. (2014) [9]	Cross-sectional clinical study	Multiple diabetes-related dermatoses (including dermopathy)	750	Cutaneous findings correlated with extracutaneous complications such as microangiopathy and neuropathy
Boulton et al. (1988) [11]	Clinicopathologic study	Necrobiosis lipoidica	35	Linked to peripheral vascular disease and chronic inflammatory changes
Rho et al. (1998) [19]	Observational study	Scleredema diabeticorum	44	Associated with poor glycemic control and metabolic syndrome
El Fekih et al. (2009) [13]	Case series	Bullosis diabeticorum	10	Occurs in the context of advanced microangiopathy and neuropathy
Zhao & Li (2023) [14]	Case report + review	Eruptive xanthomas	—	Clinical marker of severe hypertriglyceridemia and increased cardiovascular risk

**Table 2 jcm-14-07719-t002:** Summary of key clinical aspects of diabetic dermopathies and their associations with diabetes mellitus.

Condition	Diabetes Association	Clinical Presentation	Differential Diagnosis	Recommended Treatment
Diabetic dermopathy	More frequent in Type 2 and in long-standing diabetes; correlates with microangiopathy	Round or oval, atrophic, brownish macules on the pretibial area (“shin spots”), usually asymptomatic	Post-inflammatory hyperpigmentation, stasis dermatitis, lichen planus pigmentosus	No specific therapy; optimize glycemic control; emollients; monitor for microvascular complications
Necrobiosis lipoidica	Occurs in both Type 1 and Type 2; some studies suggest Type 1 > Type 2	Yellow-brown plaques with central atrophy, telangiectasias, and possible ulceration, typically on shins	Granuloma annulare, sarcoidosis, lupus panniculitis, stasis dermatitis	Glycemic optimization; topical/intralesional corticosteroids; pentoxifylline; phototherapy; wound care for ulcers
Scleredema diabeticorum	Strongly associated with Type 2, obesity, insulin resistance, poor control	Non-pitting induration and thickening of posterior neck, upper back, shoulders	Scleroderma, scleromyxedema, morphea, myxedema	Improved glycemic control; phototherapy (UVA-1, PUVA); physiotherapy; limited benefit from systemic therapy
Bullosis diabeticorum	Rare, mostly in Type 1 or long-standing Type 2 with microangiopathy	Sudden, spontaneous, non-inflammatory bullae on acral areas (feet, hands)	Bullous pemphigoid, epidermolysis bullosa acquisita, porphyria cutanea tarda	Self-limited; protect lesions; prevent secondary infection; glycemic optimization
Eruptive xanthomas	More frequent in Type 2 with severe hypertriglyceridemia (>2000 mg/dL)	Crops of yellow-red papules on extensor surfaces, buttocks, back; may be pruritic	Xanthoma disseminatum, lichen planus, molluscum contagiosum	Urgent lipid-lowering therapy (fibrate, omega-3, insulin if severe); strict diet; manage pancreatitis risk

**Table 3 jcm-14-07719-t003:** Comparative overview of diabetic dermopathies: prevalence, systemic associations, level of evidence, and key inconsistencies/negative findings.

Dermopathy Type	Approx. Prevalence in DM	Predominant Systemic Correlations	Level of Evidence	Inconsistencies/Negative Findings
Diabetic dermopathy	30–50% (esp. long-standing T2DM)	Microvascular: retinopathy, nephropathy, neuropathy	**Moderate** (multiple cross-sectional cohorts)	Some studies lose significance after adjustment for diabetes duration/age; heterogeneity in clinical vs. histologic criteria; occasional null associations with nephropathy when confounders included.
Necrobiosis lipoidica	0.3–1.2%	Macrovascular: PAD, CAD; ulcer risk	**Low–Moderate** (small series; pathophysiologic plausibility)	Inconsistent links with glycemic control and CAD across cohorts; diagnostic variability (clinical vs. biopsy) limits comparability.
Scleredema diabeticorum	up to ~2.5% in obese DM	Metabolic syndrome, insulin resistance; LV hypertrophy/diastolic dysfunction	**Low–Moderate** (cross-sectional, single-center)	Small samples; lack of longitudinal data; cardiac associations not uniformly reproduced; potential selection bias in specialty clinics.
Bullosis diabeticorum	<0.5%	Advanced microangiopathy; poor glycemic control	**Low** (case series)	Rarity → limited external validity; no adjusted analyses; causality uncertain.
Eruptive xanthomas	Rare (often with TG > 2000 mg/dL)	Severe dyslipidemia; pancreatitis; increase CV risk	**Moderate** (strong pathophysiology; limited outcomes data)	Strong association with hypertriglyceridemia but few prospective outcome studies; confounding by lipid disorders common.

Qualitative level-of-evidence grading (narrative): Moderate = multiple consistent observational studies across settings; Low–Moderate = several studies with notable limitations/heterogeneity; Low = case series/reports or small single-center studies with high risk of bias. Ratings reflect consistency and study design, not a formal risk-of-bias tool.

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
