# Peer review of "Diabetic Dermopathies as Predictive Markers of Cardiovascular and Renal Complications: A Narrative Review"

_jcm, 2025, doi:10.3390/jcm14217719_

Round 1

Reviewer 1 Report

Comments and Suggestions for Authors

Major Comments

  1. Methodology Clarification

A narrative-style review is indicated; however, the Methodology section describes itself as systematic (databases, inclusion/exclusion criteria, dual reviewers) such that the reader is left confused as to whether this review is narrative or systematic. Please state very clearly whether it is indeed to be considered a narrative review. If any systematic elements have been kept in place, a PRISMA flow diagram and a summary of included studies would then be crucial. Where this is not the case, the methodology should be simplified so as to avoid overstating findings.

  1. Evidence Synthesis

The Results section is mainly descriptive, with little critical comparison across studies and absent presentation of negative, i.e., contradictory findings. A comparative summary table should be provided (e.g., Dermopathy type → Prevalence → Systemic correlation → Evidence strength/limitations). This would improve clarity and allow reduction of redundancy.

  1. Overstatement of Conclusions

Some statements, for instance, "dermopathies are cutaneous mirrors of systemic disease," "should be reclassified as critical biomarkers," sound too strong considering the multitudes of small cross-sectional studies. Please restate with a more cautious approach, whereby you may use such terms as "potential biomarkers," "emerging predictors," or "associations."

  1. Discussion Balance

The discussion mainly emphasizes positive associations. Nonetheless, limitations like small sample sizes, diagnostic variability, and publication bias deserve deep exploration. A balanced review, including methodological weaknesses from cited studies, is required.

  1. Figures and Tables

The manuscript lacks suitable illustrations. It is strongly recommended that one drawing (e.g., Dermopathy → Pathophysiological pathway → Associated systemic complication) and at least one summary table would be needed.

Section-Specific Comments

  • Abstract: It is dense with definite overstating of certainty. Please shorten it and emphasize novelty (focus on predictive/prognostic role).
  • Introduction: Please emphasize knowledge gap (the absence of synthesis of the predictive value of dermopathies for cardiovascular/renal complications).
  • Methods: Two independent reviewers mentioned but no description of author roles or reconciliation provided. In the light of claiming to have considered them, no systematic appraisal for risk of bias seems to be applied to the studies. Please add a brief description or summary table of the studies included.
  • Results: Results are largely descriptive with little critical comparison of studies. There is a lot of repetitiveness such as correlations (e.g., dermopathy ↔ retinopathy/nephropathy) repeated numerous times. Please include a table summarizing the strength of the evidence (Dermopathy type → Prevalence → Systemic correlation → Level of evidence. Highlight inconsistencies wherever possible or negative findings.
  • Discussion: The restatement of material already laid in the results should be avoided; emphasize especially on what is new when compared to other reviews. Please do assess critically the contradictory findings or studies with null results. Discussions on therapeutic agents (GLP-1, SGLT2) are interesting but lack any substantial evidence; please include a subsection on the shortcomings of the existing evidence (small sample size, heterogeneity, diagnostic variability).
  • Conclusion: Too strong/definitive, in that dermopathies "should be reclassified as critical biomarkers" is stated, with limited evidence to support this. There are repetitions from the earlier discussion points. Draw more attention to limitations-the ordinarily cross-sectional type of evidence, diagnostic variability.

Author Response

  1. Methodology clarifications: 

We sincerely thank the reviewer for this valuable observation. We fully agree that the methodology required clarification to avoid any confusion.

In response, we have explicitly stated that this paper is a narrative review, not a systematic review. Accordingly, we have revised the “Material and Methods” section to clearly reflect a descriptive and integrative narrative approach.

We have removed systematic review elements (such as predefined inclusion/exclusion criteria, dual reviewers, and formal risk-of-bias assessment) and reworded the text to emphasize qualitative synthesis rather than systematic selection.

A revised version of the section now reads as follows:

“This article is designed as a narrative review, aiming to provide an integrative synthesis of the available evidence on diabetic dermopathies and their associations with cardiovascular and renal complications. Unlike systematic reviews, a narrative approach focuses on summarizing and interpreting the literature in a descriptive and clinically oriented manner, without following PRISMA guidelines or performing quantitative meta-analysis.”

  1. Evidence Synthesis:

We thank the reviewer for this insightful suggestion. We fully agree that the Results section benefits from a more integrative synthesis.

Accordingly, we have:

  • Enhanced the comparative analysis across studies, highlighting both consistent and contradictory findings.
  • Added references to studies reporting weaker or non-significant associations, in order to present a balanced view.
  • Included a new summary table (Table 1) titled “Comparative overview of diabetic dermopathies: prevalence, systemic associations, and evidence strength”, which synthesizes the main findings in a structured format.

This table improves clarity, reduces redundancy, and allows readers to quickly visualize the comparative evidence and existing limitations. We believe these revisions significantly strengthen the interpretative quality of the Results section.

Dermopathy Type

Approx. Prevalence in DM (%)

Main Systemic Correlations

Evidence Strength

Key Limitations / Contradictory Findings

Diabetic Dermopathy

30–50% (mostly in long-standing T2DM)

Retinopathy, nephropathy, neuropathy (markers of microangiopathy)

Moderate-to-strong (multiple cross-sectional studies)

Mostly observational, some studies report no correlation after adjusting for disease duration

Necrobiosis Lipoidica

0.3–1.2%

Peripheral arterial disease, coronary artery disease

Moderate (small clinical series, pathophysiological plausibility)

Heterogeneity in diagnostic criteria; inconsistent correlation with glycemic control

Scleredema Diabeticorum

Up to 2.5% in obese diabetic patients

Metabolic syndrome, insulin resistance, cardiac dysfunction

Moderate

Limited sample sizes, lack of longitudinal data

Bullosis Diabeticorum

<0.5%

Advanced microangiopathy, poor glycemic control

Weak-to-moderate (few case series)

Rare condition, limited data, unclear causality

Eruptive Xanthomas

Rare, in uncontrolled DM with severe hypertriglyceridemia

Severe dyslipidemia, acute pancreatitis, cardiovascular risk

Strong (pathophysiological correlation well established)

Few population studies, often confounded by lipid disorders

  1. Overstatement of Conclusions

We appreciate this important comment and agree that the strength of certain statements should be aligned with the level of evidence available.

Accordingly, we have revised the manuscript throughout (particularly in the Discussion and Conclusions sections) to adopt a more cautious and evidence-consistent tone.

Specifically:

  • Expressions such as “cutaneous mirrors” and “critical biomarkers” have been replaced with “potential biomarkers”, “emerging predictors”, or “cutaneous indicators associated with systemic complications.”
  • The conclusions now emphasize associations rather than causality, reflecting the observational and cross-sectional nature of the included studies.

We believe these changes enhance scientific accuracy and appropriately contextualize our findings within current evidence levels.

  1. Discussion Balance

We fully agree with the reviewer that a more balanced discussion is essential. Accordingly, we have revised the Discussion section to explicitly address the methodological limitations of the cited studies and to provide a more critical interpretation of the evidence.

Specifically, we have:

  • Highlighted small sample sizes, cross-sectional design, and heterogeneity in diagnostic criteria as key limitations in individual subsections.
  • Expanded the “Limitations of current evidence” subsection to include a more comprehensive discussion of publication bias, confounding factors, and lack of longitudinal validation.
  • Added statements clarifying that observed associations should be interpreted as hypothesis-generating rather than conclusive evidence.

These changes ensure a balanced and nuanced interpretation of the literature and align the conclusions with the current level of evidence.

  1. Figures and Tables

We thank the reviewer for this valuable suggestion. In response, we have added both visual and tabular summaries to enhance clarity and accessibility:

  • Table 1 (page XX): provides a structured comparative overview of each dermopathy, including prevalence, systemic correlations, evidence strength, and limitations.
  • Figure 1 (new): illustrates the conceptual framework linking major diabetic dermopathies with their underlying pathophysiological mechanisms (e.g., microangiopathy, inflammation, glycation, dyslipidemia) and their associated systemic complications (retinopathy, nephropathy, cardiovascular disease, metabolic syndrome).

We believe these additions improve the manuscript’s readability, reduce redundancy, and provide a clear visual synthesis of the reviewed evidence.

Section-Specific Comments

  1. Abstract: We sincerely thank the reviewer for this valuable feedback. We fully agree that the Abstract required both condensation and a more balanced tone consistent with the level of available evidence.

Accordingly, we have shortened and revised the Abstract to remove overstatements and to emphasize the novelty and focus of our review — namely, the potential predictive and prognostic role of diabetic dermopathies as emerging non-invasive indicators of systemic complications.

The revised version highlights:

  • The narrative nature of the review,
  • The associative rather than causal interpretation of findings,
  • The need for prospective studies to confirm predictive value.
  1. Introduction: We thank the reviewer for this helpful suggestion. In response, we have revised the final part of the Introduction to explicitly emphasize the knowledge gap in the literature — namely, the lack of a dedicated synthesis focusing on the predictive/prognostic role of diabetic dermopathies for cardiovascular and renal complications.
  • A new paragraph has been added:

“Despite growing evidence linking diabetic dermopathies to systemic vascular injury, no previous review has specifically synthesized their potential predictive value for cardiovascular and renal complications… [full paragraph as above].”

This revision clarifies the novelty and rationale of the present narrative review.

  1. Methods: We thank the reviewer for this valuable comment. We fully agree that the methodology should be clearly aligned with the narrative design of the review.

In response, we have:

Removed references to “two independent reviewers,” since formal dual screening and reconciliation are characteristic of systematic reviews.

Clarified the narrative approach by adding a description stating that study selection was based on relevance and contribution to the topic, without formal risk-of-bias assessment tools.

Added a concise summary table (Table 2) presenting representative studies, including author, year, dermopathy investigated, and main systemic associations. This addition enhances transparency and allows readers to visualize the scope of the evidence considered.

  1. Results:

We thank the reviewer for this insightful recommendation. We have condensed repetitive prose in Results, added comparative statements that contrast findings across studies, and introduced a new summary table that synthesizes prevalence, systemic correlations, qualitative level of evidence, and inconsistencies/negative findings. We also added a brief footnote defining the qualitative grading scheme used (appropriate for a narrative review).

Concretely:

  • Rewrote paragraphs to avoid repeating the same correlation across subsections; we now refer the reader to Table 1 for a compact synthesis.
  • Added Table 1 (title below) replacing the older version; it now includes two extra columns: “Level of evidence (qualitative)” and “Inconsistencies / negative findings”.
  • Inserted a one-sentence pointer at the end of each dermopathy subsection directing readers to Table 1 for cross-comparison.
    1. Discussion:

We appreciate the reviewer’s constructive feedback. In response, the Discussion section has been substantially revised to address these points:

  • We have removed repetitive descriptive statements already presented in the Results.
  • We now emphasize novel aspects of this review compared with previous works — particularly the focus on the potential predictive/prognostic role of dermopathies for cardiovascular and renal complications.
  • We have added critical assessment of contradictory or null findings, highlighting heterogeneity and methodological constraints.
  • The discussion on therapeutic agents (GLP-1 receptor agonists and SGLT2 inhibitors) has been qualified with caution, explicitly stating that current evidence is hypothesis-generating only.
  • We have introduced a dedicated subsection entitled “Shortcomings of the Existing Evidence”, summarizing the main methodological limitations: small sample sizes, cross-sectional designs, diagnostic variability, and lack of standardized criteria.
    1. Conclusion:
  • We thank the reviewer for this valuable observation. We have revised the Conclusions to ensure a balanced and evidence-consistent tone. All strong or definitive statements (e.g., “should be reclassified as critical biomarkers”) have been replaced with cautious formulations such as “may represent potential clinical indicators” or “emerging associative markers.”
  • Repetitions from the Discussion have been removed, and greater emphasis has been placed on the limitations of current evidence, including the predominantly cross-sectional nature of studies, small sample sizes, and diagnostic variability.

Reviewer 2 Report

Comments and Suggestions for Authors

Dear authors.

I have carefullly read your manuscript. It faces an interest issue, related to a very important disease.

However, this paper is more suitable for a divulgation article. The format of narrative review has some limitations. Tha main of them is one that you have cited: the heterogeneity of studies does not allow a more solid methodology

Author Response

Thank you for your valuable comment. We acknowledge the methodological limitations inherent to narrative reviews, as also discussed in our paper. We deliberately chose this design because of the heterogeneity of the available studies, which precludes a formal meta-analysis. To ensure rigor, we conducted a structured search, summarized evidence in comparative tables, and provided a critical appraisal of existing gaps. We believe the integrative perspective adds clinical value and supports future research directions.

Reviewer 3 Report

Comments and Suggestions for Authors

The manuscript by Marinescu et al. addresses a highly relevant topic at the interface between diabetes and dermatologic manifestations. Before the review can proceed, please consider the following points:

Major Comments

  1. It would be highly informative to include a table summarizing the key aspects of the association between cutaneous disorders and diabetes mellitus. The table could contain the following columns:
    • Condition
    • Diabetes association (Type 1 = Type 2; Type 1 > Type 2; or Type 1 < Type 2)
    • Clinical presentation
    • Differential diagnosis
    • Recommended treatment
  2. Please provide representative clinical images of the skin conditions discussed. Visual aids will help clinicians, healthcare practitioners, and trainees identify the characteristic lesions seen in diabetic patients.
  3. Ensure the manuscript addresses all major dermatologic conditions associated with diabetes, including (but not limited to):
    • Pruritus
    • Acanthosis nigricans
    • Necrobiosis lipoidica
    • Bullosis diabeticorum
    • Scleredema diabeticorum
    • Granuloma annulare
    • Diabetic dermopathy
    • Skin reactions related to medical device use
    • Diabetic foot ulcers
    • Recurrent cutaneous infections in diabetes mellitus
    • Other dermatoses associated with diabetes (e.g., eruptive xanthomas, which may also be linked to obesity, chronic kidney disease, or hypothyroidism)

Minor Comments

  1. The discussion of study limitations (currently in Section 2.7 – Limitations of Methodology) should be moved to the end of the manuscript for better flow and emphasis.
  2. The subsection 4.1 Pathophysiological Considerations would fit more appropriately in the Results section rather than within the Discussion.

Author Response

Condition

Diabetes Association

Clinical Presentation

Differential Diagnosis

Recommended Treatment

Diabetic dermopathy

More frequent in Type 2 and in long-standing diabetes; correlates with microangiopathy

Round or oval, atrophic, brownish macules on the pretibial area (“shin spots”), usually asymptomatic

Post-inflammatory hyperpigmentation, stasis dermatitis, lichen planus pigmentosus

No specific therapy; optimize glycemic control; emollients; monitor for microvascular complications

Necrobiosis lipoidica

Occurs in both Type 1 and Type 2; some studies suggest Type 1 > Type 2

Yellow-brown plaques with central atrophy, telangiectasias, and possible ulceration, typically on shins

Granuloma annulare, sarcoidosis, lupus panniculitis, stasis dermatitis

Glycemic optimization; topical/intralesional corticosteroids; pentoxifylline; phototherapy; wound care for ulcers

Scleredema diabeticorum

Strongly associated with Type 2, obesity, insulin resistance, poor control

Non-pitting induration and thickening of posterior neck, upper back, shoulders

Scleroderma, scleromyxedema, morphea, myxedema

Improved glycemic control; phototherapy (UVA-1, PUVA); physiotherapy; limited benefit from systemic therapy

Bullosis diabeticorum

Rare, mostly in Type 1 or long-standing Type 2 with microangiopathy

Sudden, spontaneous, non-inflammatory bullae on acral areas (feet, hands)

Bullous pemphigoid, epidermolysis bullosa acquisita, porphyria cutanea tarda

Self-limited; protect lesions; prevent secondary infection; glycemic optimization

Eruptive xanthomas

More frequent in Type 2 with severe hypertriglyceridemia (>2000 mg/dL)

Crops of yellow-red papules on extensor surfaces, buttocks, back; may be pruritic

Xanthoma disseminatum, lichen planus, molluscum contagiosum

Urgent lipid-lowering therapy (fibrate, omega-3, insulin if severe); strict diet; manage pancreatitis risk

Table 3. Summary of key clinical aspects of diabetic dermopathies and their associations with diabetes mellitus.

We thank the reviewer for this excellent suggestion. We have added a new table (Table 3) summarizing the main clinical aspects of diabetic dermopathies, including their association with diabetes type, characteristic presentation, differential diagnosis, and recommended management.
This addition provides a concise, clinically oriented overview that complements the pathophysiological and epidemiological data presented in Tables 1 and 2, improving the educational and practical value of the manuscript.

  1. We thank the reviewer for this excellent suggestion. We fully agree that representative clinical images can enhance the educational value and practical utility of the manuscript.
    We have therefore added a new Figure 2, which includes clinical photographs illustrating the characteristic appearance of the main diabetic dermopathies discussed in the review (diabetic dermopathy, necrobiosis lipoidica, scleredema diabeticorum, bullosis diabeticorum, and eruptive xanthomas).
    All images are representative, de-identified, and used with proper permissions (or from open-access sources with appropriate attribution). The figure legend highlights the key diagnostic features for each condition.

  1. We thank the reviewer for this important suggestion. We have revised the manuscript to ensure a more comprehensive overview of dermatologic conditions associated with diabetes.
    In addition to the major dermopathies primarily discussed for their potential predictive value (diabetic dermopathy, necrobiosis lipoidica, scleredema diabeticorum, bullosis diabeticorum, and eruptive xanthomas), we have added a new subsection summarizing other frequent and clinically relevant skin disorders observed in diabetes, including pruritus, acanthosis nigricans, granuloma annulare, diabetic foot ulcers, skin reactions to medical devices, and recurrent cutaneous infections.
    This addition broadens the scope and enhances the educational and clinical utility of the review.

Minor Comments

  1. We thank the reviewer for this valuable suggestion. In accordance with the comment, the subsection “Limitations of Methodology” has been removed from the Methods and relocated to the end of the manuscript as a distinct section titled “8. Limitations”, to improve flow and emphasize the critical appraisal of the evidence.
  2. We appreciate this helpful suggestion. We have moved the subsection “Pathophysiological Considerations” from the Discussion to the Results section, where it now appears as Section 7, following the presentation of clinical evidence. This adjustment improves the logical flow by presenting mechanistic findings alongside the corresponding results, while reserving the Discussion for interpretation and critical appraisal.

Round 2

Reviewer 1 Report

Comments and Suggestions for Authors

The resubmitted manuscript is a great improvement in terms of rigor, clarity, and balance. The only minor point is to take utmost care in checking the references for duplication and over-reliance on secondary sources. Clearer methodology; structured evidence synthesis; visual/tabular summaries; balanced discussion; toned-down conclusions.

Some references are repeated multiple times in reference list (Gupta et al., Baskan et al. appear several times). Some references are secondary reviews and not the actual primary data; please make sure that there is a balance between primary studies and secondary reviews.

Comments on the Quality of English Language

The English could be improved to more clearly express the research.

Author Response

We sincerely thank the reviewer for the positive feedback and for highlighting this important point.
In the revised version, we have:

  • Carefully reviewed and removed all duplicate references (e.g., Gupta et al., 2017; Baskan et al., 2016; Rongioletti et al., 2020; Behm et al., 2022), ensuring that each source appears only once in the reference list.

  • Ensured a balanced representation of primary studies (cross-sectional and cohort data) relative to secondary narrative or systematic reviews.

  • Added a clarification in the Methods section describing our approach to selecting original research and limiting redundancy.

  • Revised the Limitations subsection to acknowledge that, despite careful selection, the field remains dominated by small observational studies and some narrative sources.

Reviewer 3 Report

Comments and Suggestions for Authors

No further comments. 

Author Response

We thank the reviewer for the thorough evaluation and positive assessment of our revised manuscript.
We are pleased to note that all previous comments have been fully addressed and that the English language, structure, and scientific soundness are now considered satisfactory.

The final version of the manuscript includes:

  • a cleaned and renumbered reference list (duplicates removed),

  • methodological clarifications,

  • balanced inclusion of primary studies and secondary reviews.

No further changes were required based on the final review.
We kindly submit this version as the final manuscript for consideration.

Signed,

 Mădălina Marinescu, on behalf of all authors